# Modeling Earthen Treatments for Climate Change Effects

**Sharlot Hart [1],\*, Kara Raymond [1], C. Jason Williams [2] , William A. Rutherford [2] and Jacob DeGayner [1]**

1   US National Park Service, Southern Arizona Office, Phoenix, AZ 85712, USA; kara_raymond@nps.gov (K.R.); jacob_degayner@nps.gov (J.D.)
2   US Department of Agriculture, Agricultural Research Service, Southwest Watershed Research Center, Tucson, AZ 85719, USA; jason.williams@usda.gov (C.J.W.); austin.rutherford@usda.gov (W.A.R.)
\*   Correspondence: sharlot_hart@nps.gov; Tel.: +1-602-882-7223

**Abstract:** Adobe has been used globally for millennia. In the US Southwest, cultural heritage sites made of adobe materials have lasted hundreds of years in an arid/semi-arid environment. A common prediction across multiple climate change models, however, is that rainfall intensity will increase in the US Southwest. This increased erosivity threatens the long-term protection and preservation of these sites, and thus resource managers are faced with selecting effective conservation practices. For this reason, modeling tools are needed to predict climate change impacts and plan for adaptation strategies. Many existing strategies, including patching damaged areas, building protective caps and shelter coating walls are already commonly utilized. In this study, we modeled adobe block construction, subjected extant walls to a local 100-year return interval rainfall intensity, and tested earthen-coat-based strategies to minimize the deterioration of earthen fabric. Findings from the resultant linear models indicate that the patching of earthen architecture alone will not prevent substantial damage, while un-amended encapsulation coats and caps provide similar, and significantly greater protection than patching. The use of this model will enable local heritage resource managers to better target preservation methods for a return on investment of the material and labor costs, resulting in better preservation overall and the retention of culturally valuable resources.

**Keywords:** cultural heritage; deterioration; adobe; earthen architecture; climate change; heritage preservation; erosion

## 1. Introduction

Moisture is a major driver of damage to adobe architecture. The intensity and duration of rain events contribute to the severity of erosion and the occurrence of catastrophic collapse of earthen structures. Climate projections for the US Southwest have heightened the need to understand the relationship between rain intensity and resource damage, and to identify effective preservation methods for withstanding a range of climate futures. Climate change models for the US Southwest indicate that the frequency of high intensity rainfall is likely to increase over the next century [1–5]. Other studies have found that the frequency of extreme precipitation is already increasing in the US Southwest [6–8]. These changing precipitation patterns are likely to increase the deterioration of earthen structures [9].

The US National Park Service (NPS) preserves historic period adobe resources in a state that conveys their appearance as ruins. In general, ruins are defined as resources that no longer have their basic structural components. For adobe resources on NPS lands, many adobe ruins no longer have roofs [10]. Repairs such as patches, caps, and encapsulations are common strategies to protect unroofed and otherwise unprotected walls from damage caused by precipitation. The constituency of stabilization and repair materials is, therefore, critical for providing long-term protection at these sites. Fort Selden in the US state of New Mexico, the location of the Getty Institute's long-term study on the productivity of earthen shelter coat amendments [11], is a prime example of attempts to test materials on adobe ruins. While many NPS sites in the US have test walls, they are built to test

treatment efficacy in ambient weather conditions. This is the case at both Pecos National Historical Park in northern New Mexico [12] and on site at Tumacácori National Historical Park (TUMA) [13] in southern Arizona.

The experiment and results presented here grew out of a pilot study modeled on extant adobe walls at the site of Mission Los Santos Angeles de Guevavi, a part of TUMA [9], but was physically conducted in Tucson, Arizona, where the logistics of modeling could be streamlined. While the current work is centered primarily at sites in central and southern Arizona, US, damage from increasing precipitation intensity is expected to impact adobe resources throughout the US Southwest. To systematically investigate the impact of precipitation intensity and duration on earthen architecture, 20 adobe test walls were constructed using materials and methods consistent with the historic fabric comprising adobe buildings at TUMA. Test wall materials and construction are described in detail in the aforementioned Hart et al. [9] companion study. The bricks and walls were constructed during two training sessions for NPS cultural resource personnel and their cooperators in August and October 2018. The training sessions were led by three instructors experienced in both masonry and adobe construction and in preservation in southern Arizona.

No preservation treatments or amendments were applied to the test walls. In November 2018, a rainfall simulator was used to apply one of four high-intensity rain treatments to each test wall, based on the return interval for 1-year, 25-year, and 100-year storms as estimated for the TUMA weather station USC000028865 [9,14]. The 30 min rainfall treatments were (1) control: no rainfall, (2) 1-year storm: 3.6 cm h$^{-1}$, (3) 25-year storm: 8.5 cm h$^{-1}$, and (4) 100-year storm: 10.6 cm h$^{-1}$. This study found that the 30 min, 100-year storm caused a mean wall material loss of 5.64% and affected a mean area of 8790 cm$^2$ of the wall surface area. This earlier study found that an increasing rainfall intensity, as predicted in the climate change models mentioned above, will cause increased rates of erosion in unprotected adobe block construction.

This valuable data on the impact of the rainfall amount and intensity on bare adobe will allow cultural resource managers to anticipate and prepare for how a range of rain intensities can impact bare adobe. Since many adobe resources on NPS land already have protection strategies in place to mitigate the erosive effects of increasing rainfall intensity; the purpose of the current study is to evaluate three common protection measures using unamended earthen treatment options—patches, caps, and encapsulation/shelter coats. To address the goal, we (1) randomly applied patch, cap, and encapsulation/shelter coats to individual adobe test walls, (2) applied a local 100-year rainfall intensity for 30 min to each wall using rainfall simulation methods, and (3) used terrestrial laser scanning methods to quantify the wall deterioration.

## 2. Materials and Methods

Since their original construction, the test walls have experienced variable amounts of erosion in response to both ambient weather and the applied rainfall events [9]. As detailed above, in November 2018, rainfall simulators were used to apply one of four high-intensity rain treatments to each test wall. Additionally in that study, subsequent additional low-intensity rainfall simulation experiments were conducted in May–June 2019 on a subset of walls applying 0.97 cm h$^{-1}$ of rainfall in either a single event (1 event, 240 min) or two events (2 event, 80 min and 160 min events separated by a 48 h hiatus) to assess the effects of prolonged low-intensity rainfall on wall degradation. Untreated walls (control treatment) for the low-intensity simulations utilized the same original five control test walls from the previous high-intensity rainfall simulations. The walls received no preservation treatments or amendments after the high- or low-intensity experiments. After these two experiments, all the walls were primarily exposed to ambient weather conditions. By Autumn of 2022, the walls' erosional state had visually appeared as "melted adobe ruins" (Figure 1).

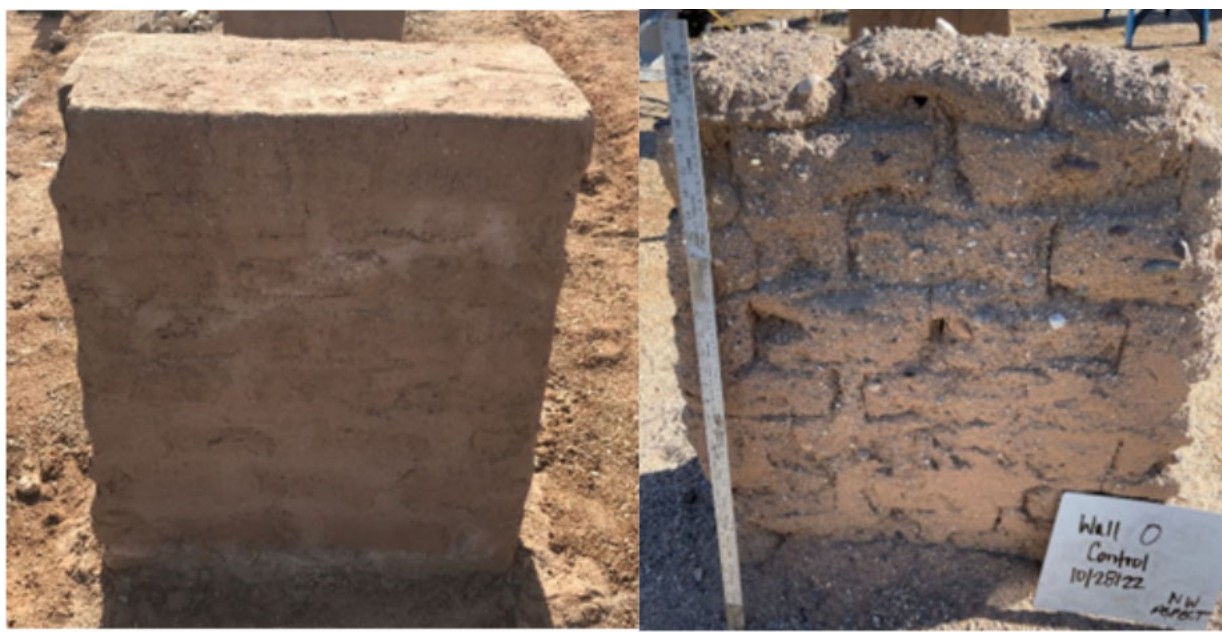

**Figure 1.** Wall O when newly constructed in 2018 (**left**) and before rain simulation in 2022 (**right**).

While the previous study [9] used rainfall intensities to correlate the precipitation and material loss, this study sought to investigate the effectiveness of common treatments under one future rainfall intensity. In this case, all the walls, including the test locations originally classified as control walls, were subjected to 100-year rainfall intensity (10.6 cm $h^{-1}$) for 30 min. This scenario was intended to test the effectiveness of three common types of treatments. Each treatment used unamended earthen material, consisting of the same locally-sourced soil with clay content and sand as the previous study [9]. Unamended earthen material is defined as having no chemical additives such as polymer adhesives to help with water shedding. As each locally-procured soil will have distinct characteristics, there is no one formula for mixing the clay, sand, and water, and studies and guides on adobe give a range of 0 to 30% clay content [15–18]. However, a certain consistency is desirable and was reached, namely, one where the wet material stayed adhered to a trowel when inverted and spread easily on the wall surfaces. All the treatments were hand applied with trowels. The treatments were applied to 5 walls each in the following configurations (see also Figure 2):

1.  Control: no unamended earthen material added.
2.  Patch: earthen material used to fill in voids and cracks, in some cases cobbling/rajuelas were applied.
3.  Cap: earthen material in a wet-plaster consistency applied over the tops of the walls and vertically down the four faces of the walls to 10 cm; depth/thickness of plaster did not exceed 1.25 cm, excluding locations where patching was performed. Patching was performed prior to capping.
4.  Encapsulate (encapsulation/shelter coat): earthen material in a plaster consistency applied to the entire wall surface; that is, a cap plus coating the walls to the ground surface. Depth/thickness did not exceed 1.25 cm, excluding locations where patching was performed. Patching was performed prior to encapsulation.

Walls were randomly assigned to treatments. Because the walls had variable amounts of erosion from previous rain simulations and impacts from ambient rainfall, a one-way analysis of variance (ANOVA) was performed to identify any differences in the overall degradation between the randomized treatment groups. This analysis found that the treatment groups chosen immediately prior to the treatment application in 2022 were not significantly different from each other relative to their total volume loss ($p = 0.79$) and affected surface area ($p = 0.95$).

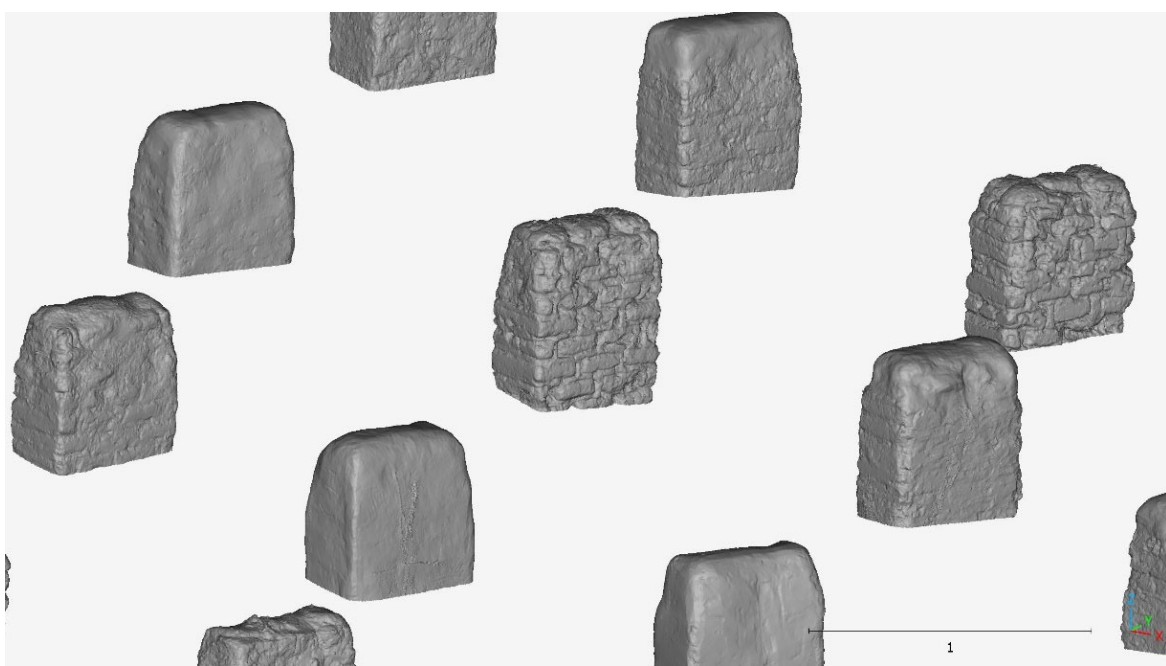

**Figure 2.** Lidar scan/wall models of treated walls prior to 100-year (10.6 cm h$^{-1}$) rainfall event application.

The mean amount of unamended material added per treatment is summarized in Table 1. While the control treatment did not receive any added material, the LiDAR scans indicate the control walls lost a mean of 577 cm$^3$ of original fabric prior to this study. This is primarily due to the physical disturbance from covering the walls with tarps. Tarps were applied to the walls one week prior to the rainfall simulations to protect them from forecasted rain immediately prior to the experiments. Some gravel-sized rocks were dislodged from the top of the walls where the surrounding adobe material had previously eroded. Wind likely caused the secured tarps to abrade the walls, eroding fines and causing material loss. All the walls were tarped, and similar losses likely occurred at the non-control walls but could not be quantified. The high standard deviation for the patch and encapsulate methods is due to the variable degradation of the walls prior to applying the treatment (Table 1). Comparatively, the cap application was much more uniform.

**Table 1.** Mean (standard deviation) volume (cm$^3$) of material added for each treatment and percent added relative to pre-treatment.

| Treatment | Mean Added Material (cm$^3$) | | Mean Added Material Relative to the Pre-Treatment Volume (%) | |
|---|---|---|---|---|
| Control | −577 | (SD 228) | −0.253 | (SD 0.107) |
| Patch | 8043 | (SD 4265) | 4.06 | (SD 2.08) |
| Cap | 14,015 | (SD 1760) | 6.61 | (SD 0.515) |
| Encapsulate | 27,157 | (SD 17,287) | 14.3 | (SD 4.59) |

### 2.1. Rainfall Simulator

Rainfall simulations were conducted over the period of 25 October–1 November 2022 with consistent daily ambient weather conditions without natural rainfall. Rainfall simulations employed the same portable single-nozzle, Meyer and Harmon-type, oscillating-arm rainfall simulator [19–21] used in earlier high- and low-intensity rainfall studies on the test walls [9]. The rainfall simulator (Figure 3) was fitted with a VeeJet 80–100 nozzle raised 3 m above the ground surface and supplied with water pressurized at 41 N m$^{-2}$. The raindrop size and kinetic energy of the simulated rainfall was within 1 mm and

70 kJ ha$^{-1}$ mm$^{-1}$, respectively, of the values reported for natural convective rainfall [19,22,23]. Tarps were applied to the rainfall simulator prior to the experiments to prevent wind effects on rainfall and to ensure consistency in the rainfall application rate. Specifically, the simulator configuration described above applies rainfall with a kinetic energy at the ground surface of approximately 200 kJ ha$^{-1}$ mm$^{-1}$ and about 2 mm average drop size [19]. The simulator was calibrated multiple times daily for the target intensity (10.6 cm h$^{-1}$) by simulating rainfall over a calibration pan for five minutes [20,21]. The calibrations resulted in an average application rate of 10.57 cm h$^{-1}$ (applied for a 30 min duration) with a standard deviation of 0.01 cm h$^{-1}$ across all the simulations. All the calibration runs and wall simulations were controlled for wind by tarping around the simulator and respective test wall.

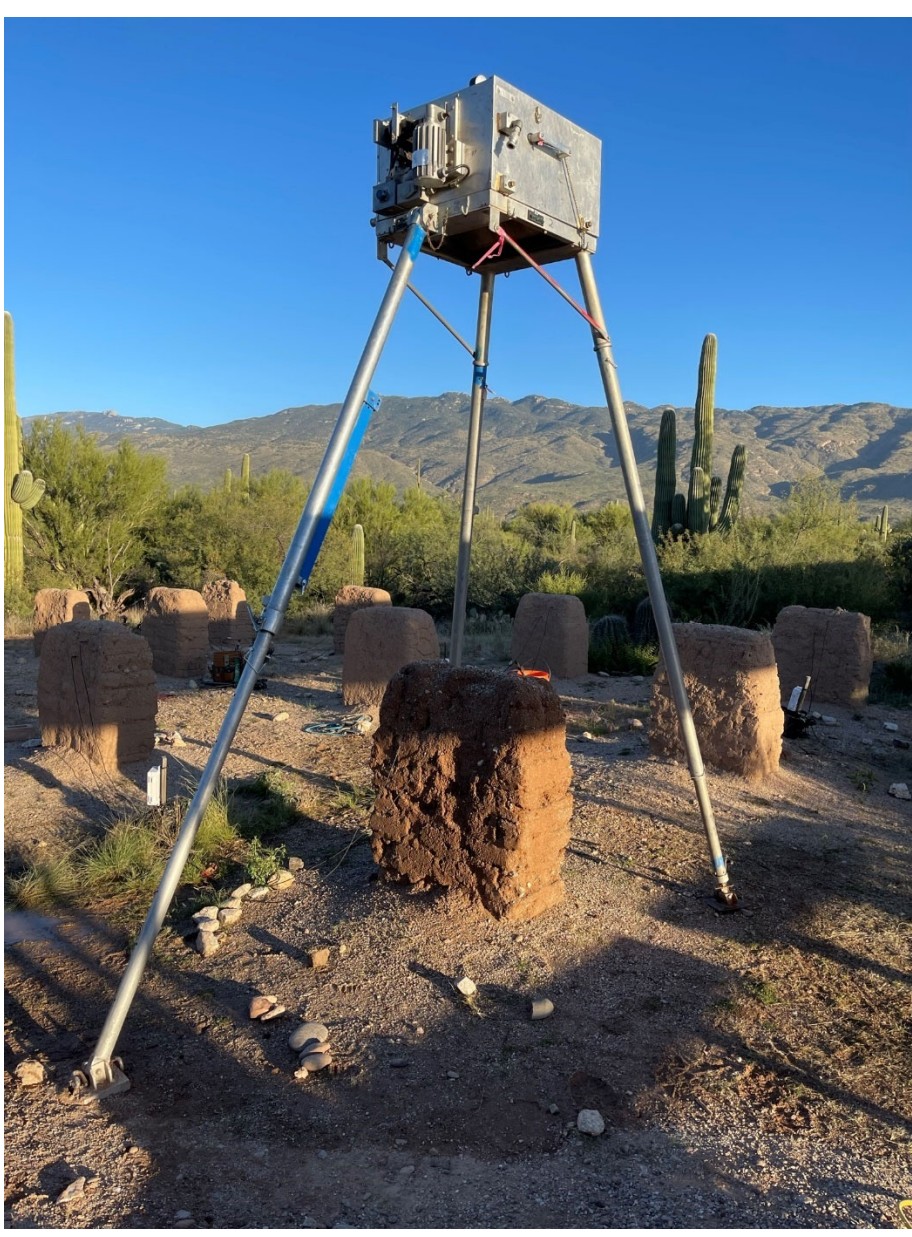

**Figure 3.** Rainfall simulator with tarps removed, October 2022.

*2.2. Lidar Application*

A Surphaser® Model 10 was used to measure the test walls in 3D space, with a margin of error likely to remain under one millimeter for distances within 15 m of the scanner [9,24]. In addition, the same two methods of registration (i.e., spherical targets and permanent control points) used in Hart et al. [9] were again employed to improve accuracy.

We recorded test walls in three scanning epochs, each consisting of approximately 30 scans, over a 42-day period. The epochs consist of:

- Epoch IX: immediately prior to treatment application; 6 September 2022,
- Epoch X: after treatment application and before the rain simulation; 28 September 2022 (Figure 2),
- Epoch XI: after the rain simulation; 3 November 2022.

Measured and adjusted locations of the control points from each scanning epoch were compared to the global mean location derived from all three epochs, resulting in a mean target residual distance of 0.54 mm.

Post-processing methodologies remained the same as in previous experiments [9], including processing and registration in FARO Scene software (SCENE, version 2021.1), and then co-registering with the other epochs using the five permanent control points. We again defined the reference planes above mass wasting build-up levels and removed all the surfaces below.

Wall-degradation metrics (i.e., material loss for treatment and original fabric, and material loss, affected surface area, and maximum recession distance for original fabric only) were calculated for each wall using the before and after lidar scans. The material (%) and volume losses (cm$^3$) were defined as the relative difference between pre- and post-treatment wall volumes. Scanning epochs were transformed in single point clouds and converted to polygonal meshes in the same way as the previous study, but with Geomagic Wrap (v.2021). A hole-filling operation was again used and the volumes calculated.

The affected surface area was defined as wall surfaces that exhibited a deviation greater than or equal to 2 mm from the previous epoch. We calculated this error threshold in previous experiments by measuring deviations on non-treated control walls, yielding a confidence interval of approximately 99%. We re-used this value due to the identical methodology used in this study. Distances were computed using the Cloud-to-Mesh Distance tool in CloudCompare (v.2.11.2) to generate a scalar field, calculating a signed distance value to each mesh facet. Negative values indicate surfaces on a compared model below the corresponding area of the reference model. Conversely, positive values indicate areas above the reference model. Model facets representing positive and negative values in excess of 2 mm in either direction were isolated for each wall model (Figure 4). Thus, surface areas exhibiting negative values on Epoch IX compared to Epoch XI represent the areas where wall surfaces have eroded past treatment material into the "original" fabric (Figure 5). Positive values indicate areas where preservation material was still present. The maximum recession distance records the greatest negative value present for each test wall.

Material losses for the treatment and original fabric were derived as the relative difference between the Epoch X and XI wall volumes. Because of the material addition between scanning Epochs IX and X, direct measurements of the wall model volume could not be used to assess the amount of original fabric material loss. Thus, the material loss of original fabric only was calculated as the affected surface area measurements referenced above, multiplied by the signed distances to generate an indirect measurement of loss.

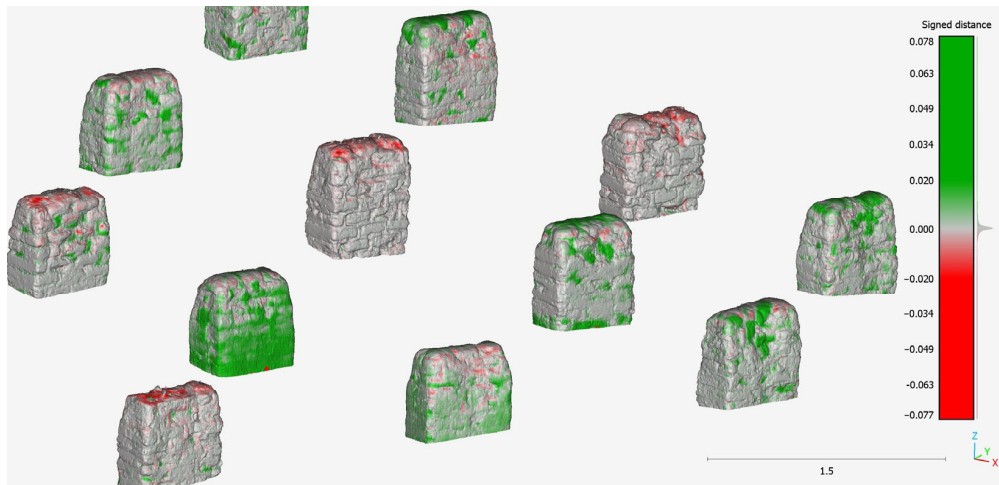

**Figure 4.** Epoch XI walls compared to Epoch IX showing positive signed areas representing additions (green) and negative signed areas representing loss (red).

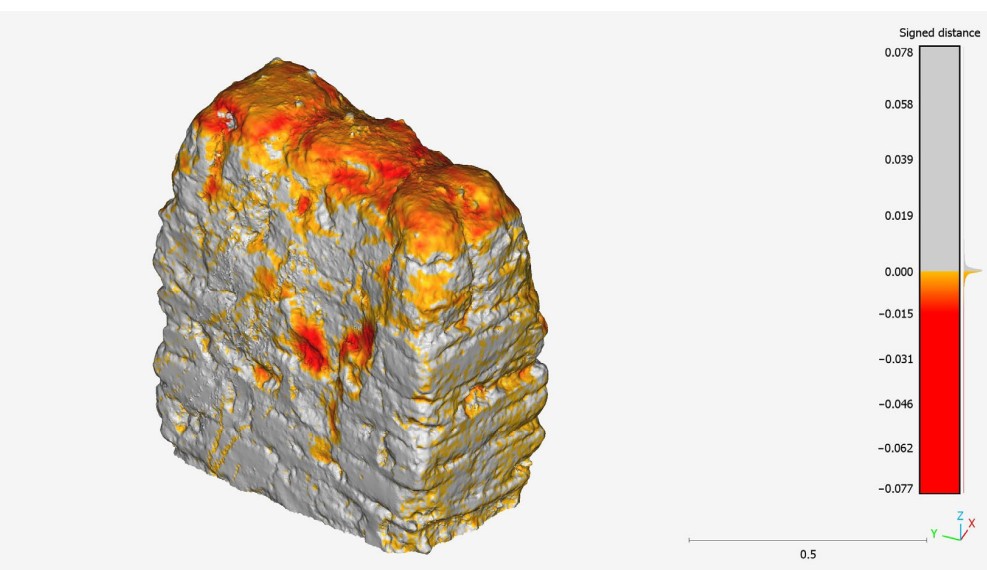

**Figure 5.** Affected area and negative signed values (in orange-red) used to calculate original material loss on Wall C.

*2.3. Data Analysis*

We computed a linear model by treatment (i.e., control, patch, cap, and encapsulate) for the four metrics: loss of original and treatment material, surface area receded past original material, impact on original material, and maximum deviation from original material. The metrics were analyzed using a one-way analysis of ANOVA and Tukey's post-hoc test.

We used the R programming language with RStudio (v.4.2.0) for the analyses: stats (v.4.2.0) for linear models, emmeans (v.1.8.3) to estimate the marginal means, car (v.3.1-1) for the type-II ANOVA (F-tests for linear models), and multcomp (v.1.4-20) to generate group letters of Tukey pairwise comparisons.

**3. Results**

The simulated rain event caused the patch, cap and encapsulate treatments to lose a mean of 3.37–3.75% of their total volume, including the original fabric and added treatment material (Table 2, Figure 6A). There was no significant difference between the three preservation treatments. Significantly less material eroded away in the control treatment, with a mean of 0.77% of the total volume, which was only comprised of original fabric.

**Table 2.** Wall treatments and erosion metrics.

| Wall | Treatment | Material Loss of Original and Treatment Fabric (%) (cm³) | Affected Surface Area of Original Material (%) (cm²) | Material Loss of Original Fabric (cm³) | Maximum Recession Distance from Original Material (cm) |
|---|---|---|---|---|---|
| C | Control | 1.03 (2432) | 15.85 (3718) | 2880 | 4.11 |
| M | Control | 0.70 (1681) | 13.76 (3250) | 2109 | 3.26 |
| O | Control | 0.61 (1059) | 12.93 (2427) | 1653 | 2.70 |
| P | Control | 0.73 (1726) | 13.93 (3177) | 2114 | 3.92 |
| R | Control | 0.79 (1753) | 15.84 (3539) | 2298 | 3.27 |
| A | Patch | 2.56 (5240) | 18.32 (3801) | 3555 | 4.43 |
| B | Patch | 3.24 (6330) | 13.16 (2517) | 1416 | 3.73 |
| H | Patch | 4.08 (8106) | 13.32 (2814) | 2055 | 4.33 |
| I | Patch | 3.34 (6154) | 13.52 (2590) | 2193 | 3.25 |
| T | Patch | 3.60 (7779) | 6.00 (1256) | 796 | 2.97 |
| D | Cap | 4.35 (9171) | 2.88 (564) | 829 | 6.11 |
| K | Cap | 3.35 (6970) | 5.00 (1055) | 852 | 2.70 |
| L | Cap | 2.61 (6198) | 3.04 (677) | 328 | 2.68 |
| Q | Cap | 3.53 (6946) | 4.24 (833) | 482 | 3.04 |
| S | Cap | 3.15 (7514) | 6.20 (1412) | 865 | 3.61 |
| E | Encapsulate | 3.16 (6666) | 2.56 (487) | 286 | 2.95 |
| F | Encapsulate | 3.95 (7594) | 11.97 (2210) | 1264 | 2.95 |
| G | Encapsulate | 4.27 (9191) | 4.19 (842) | 546 | 3.11 |
| J | Encapsulate | 3.30 (7317) | 3.45 (682) | 658 | 8.79 |
| N | Encapsulate | 4.07 (8316) | 7.41 (1524) | 936 | 2.65 |

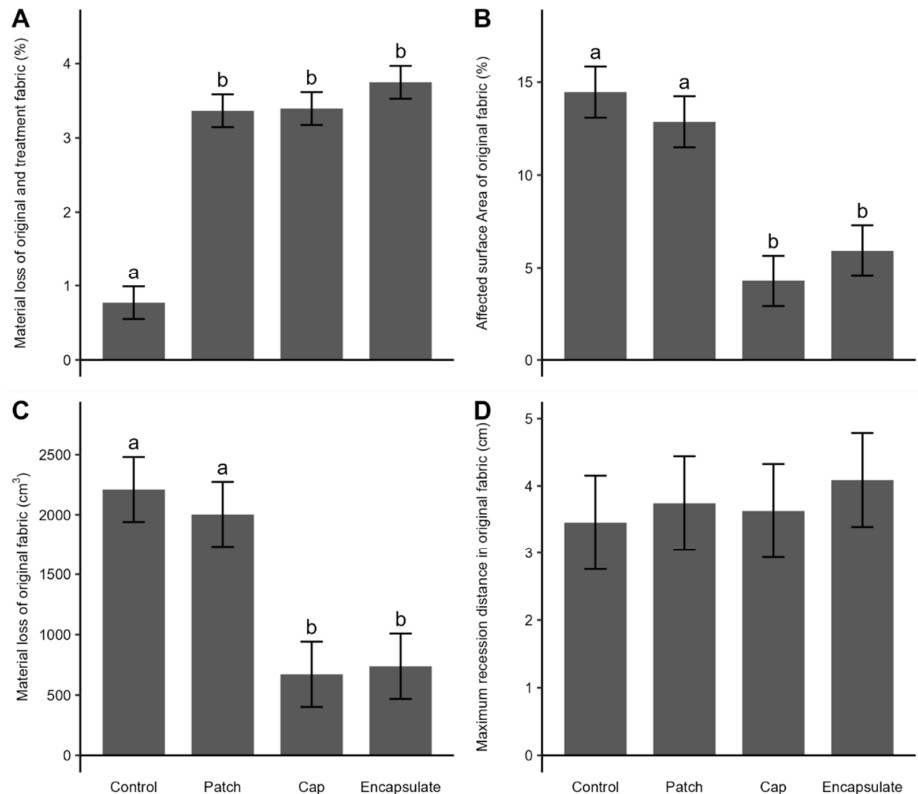

**Figure 6.** Estimated marginal mean (±SE) for erosion and moisture metrics by preservation treatments. Lowercase letters across treatments within a metric (**A–D**) indicate statistically different groups based on post-hoc pairwise comparisons, Tukey method ($p \leq 0.05$).

The affected surface area and material loss metrics related to the original walls show where the erosion went past the treatment material and into the original wall fabric. These metrics indicate that the control and patch treatments resulted in significantly greater degradation of the original fabric than the cap and encapsulate treatments (Table 2, Figure 6B,C). The simulated rain event caused a mean of 14.5% of the surface area of the pretreatment control and 12.9% of the patch walls to recede $\geq 2$ mm; however, the two treatments were statistically the same. Erosion of the surface area was significantly less severe in the cap and encapsulate treatments, with a mean of 4.27% and 5.92%, respectively. The material loss of original fabric indicates mean losses of 2210 and 2000 cm$^3$ for the control and patch treatments, respectively. The cap and encapsulate treatments showed significantly less material loss on the original fabric, with approximately one-third of that for the control and patch treatments.

The type of preservation treatment used was not a strong predictor for the maximum recession distance into the original fabric (Figure 6D). The mean maximum recession distance was lowest in the control treatment and highest in the encapsulate treatment. However, the F-test (for variability between the group means) indicated that none of the treatments were significantly different, and thus, post-hoc tests were not conducted (Table 3).

**Table 3.** Analysis of variance results for erosion metrics.

| Parameter | Material Loss of Original and Treatment Fabric (%) (cm$^3$) | Affected Surface Area of Original Material (%) (cm$^2$) | Material Loss of Original Fabric (cm$^3$) | Maximum Recession Distance from Original Material (cm) |
|---|---|---|---|---|
| F-test | 39.0 (41.4) | 14.9 (13.4) | 9.07 | 0.147 |
| *p*-value | <0.001 (<0.001) | <0.001 (<0.001) | <0.001 | 0.93 |

## 4. Discussion

### 4.1. Return on Investment for Sacrificial Materials

The preservation treatments in this study are all designed to be sacrificial treatments on top of the historic adobe, providing a protective layer that will erode before the historic adobe underneath. The earthen mixture used is similar to mortar and has a higher erodibility due to its higher content of clay and other fine particles compared to the adobe block fabric [9]. This study supports what historic preservationists have observed in the field, namely, that "the cap is the life of the wall" [25]. Statistically, there is no difference in the material affected, across all metrics, between a cap and an encapsulation coat treatment. There is, however, a statistical difference regarding the affected surface area and material loss on the original fabric affected between solely patching and either placing a sacrificial cap or encapsulation coat.

Each agency or entity that maintains historic architecture somehow defines its maintenance cycle, and the associated costs. Since those terminologies and definitions differ, here we borrow the concept of "return on investment" (ROI) from the business world, where ROI answers the question if there is a net benefit to investing in one of these treatments, e.g., [26]. In a general sense, it is easy to evaluate the staff time and materials needed for each of the three treatments: patching at the lower end of investment, capping in the middle, and full encapsulation at the upper end of the needed investment. For example, during the workshop used to apply the treatments to the walls in this study, a skilled practitioner was able to patch two walls, in the time it took another skilled practitioner to fully encapsulate one wall. This model not only validates the idea that an applied cap is vital to the preservation of original earthen fabric, but also provides heritage practitioners and managers with data to decide when and where to focus staff energy and funding.

For example, given the projected increased rainfall intensity discussed earlier, if a heritage area or other adobe site has full-time preservation practitioners on staff, our models

show that their time investment to re-cap or re-encapsulate multiple times within a year may be a positive ROI. A site without full-time practitioners on staff, however, likely does not have the capacity to make the same choice and will need to look for other treatment options (e.g., amended earthen coats) to extend their cyclic preservation maintenance. This is certainly the case at another NPS heritage site in southern Arizona, Fort Bowie National Historical Park, where Porter [27] (p. 13) reported: "Sheltering with unamended earthen plaster requires a substantial commitment to maintenance of the shelters following each heavy rainfall; this is far beyond the current capacity of Fort Bowie, where one staff person is responsible for maintenance of park facilities as well as preservation maintenance of the ruins".

*4.2. Armoring of Original Fabric*

The results of the present study and high-intensity rainfall treatments on these test walls in 2018 suggest the walls became more armored, increasing their resistance to erosion [9]. In the 2018 study, a treatment group of the newly constructed bare adobe walls received a 30 min applied rainfall with an intensity of 10.6 cm h$^{-1}$, the same as in the study presented here; however, the control walls (n = 5) in this present study were not new, as they had been exposed to four years of ambient weather and then simulated rain events. We found that the erosion results were substantially different for these weathered walls, compared to the new walls that received the same simulated rainfall event (n = 5) (Table 4). The mean material loss on the new walls was over eight times that of the degraded walls. The mean affected surface area of the walls in the 2018 study was twice as much as the affected surface area of the weathered walls. This suggests that the original fabric became more resistant to intense rainfall over the period of four years.

**Table 4.** Mean (and standard deviation) erosion metrics (cm$^3$ and %) for new and degraded walls, both exposed to a 30 min, 10.6 h$^{-1}$ simulated rain event.

| Wall Age | Material Loss | | Affected Surface Area | |
|---|---|---|---|---|
| | cm$^3$ | % | cm$^2$ | % |
| New walls | 14,400 | 5.64 | 6440 | 28.9 |
| (2018 study) | (SD 2560) | (SD 1.00) | (SD 432) | (SD 3.29) |
| Weathered walls | 1730 | 0.764 | 3220 | 14.5 |
| (present study) | (SD 435) | (SD 0.139) | (SD 443) | (SD 1.18) |

Progressive armoring of the weathered adobe walls may account for some of the differences in the rate of erosion. Soil armoring may be accomplished through natural or anthropogenic means [28,29], but in this case we refer to the ability of larger particles including gravel and rock to reduce erosion [30,31]. The newly constructed walls had a smooth finish and flat top (Figure 1). Through successive simulated and ambient rain events, wind erosion, and abrasion from periodic tarping (for short periods immediately prior to experiments), the surface texture and shape of the walls changed drastically; fine particles were transported off the walls leaving the larger sand and gravel particles, the tops became more convex, and the mortar eroded faster than the bricks. Protruding sand and gravel can absorb the impact of raindrops and slow runoff; thus, reducing erosion. The convex top prevents the pooling of water, which may reduce the concentrated pour over and rilling on wall faces. Some runoff may route along the incised mortar lines, which may slow the runoff at the horizontal sections. Generally, these results support the observation that the condition of the top of a wall is a key determinant to the preservation of the wall, and even a natural armored adobe wall can provide some protection. Protection is relative, however; while the weathered walls here lost a fraction of the original fabric compared to the new walls under the same storm conditions, 1730 cm$^3$ and 14,400 cm$^3$, respectively, that smaller loss still constitutes a heritage conservation issue (Table 4).

*4.3. Wind-Driven Rain*

This study omitted the effects of wind on rainfall and wall degradation and material loss. This was intended to focus more specifically on the effects of rainfall intensity and to ensure consistency in the rainfall application rate across treatments. As such, the wall degradation and losses as quantified here may be conservative relative to responses under wind-driven rain [32,33]. Wind can strongly influence rainfall intensity and the impact velocity and direction of raindrops [33]. The magnitude of these effects varies with the horizontal wind velocity [34]. The respective relationships are complex [32–35] and are outside the scope of the current study. Substantive discussion on the effects of wind-driven rain on the degradation of historical building materials is provided by Blocken and Carmeliet [36] and Erkal et al. [35].

**5. Conclusions**

This study attests to the real-world application of models in heritage work. There is often a disconnect between the offices and staff who have the capacity to perform modeling experiments and those who are on the ground, faced with everything from hairline cracks to imminent collapse. This study seeks to test common strategies used to minimize deterioration to adobe ruins—and one that comes from the field. While all the treatments tested provided a level of protection, capping walls with unamended earthen material will provide some protection in light of increasing storm intensity. It is also less time and material intensive than an application of an encapsulation coat.

However, the data provided by the models do not conclude that either the cap or the encapsulation treatments "saved" the walls from increasing rainfall intensity. Indeed, while these treatments were statistically different from the control and patch treatments, they still had measurable impacts to the original fabric, after only a 30 min storm at the 100-year intensity. The application cycles for unamended earthen caps would need to be far shorter than they currently are at many NPS sites.

As such, we intend to use these same models to continue the experiments, testing different additives to earthen material and other sacrificial coat options. Finally, due to the unexpected findings regarding differences in the rate of erosion of new walls to weathered walls, further study is needed to refine our models, and to investigate soil armoring on three-dimensional objects (walls) to maximize the return on investment in preserving these nationally important heritage sites.

**Author Contributions:** Conceptualization, S.H.; methodology, S.H., K.R., J.D. and C.J.W.; formal analysis, J.D. and K.R.; investigation, validation, resources, data curation, writing—original draft preparation, writing—review and editing, S.H., K.R., C.J.W., W.A.R. and J.D. All authors have read and agreed to the published version of the manuscript.

**Funding:** This research received no external funding.

**Data Availability Statement:** The data are available as figures in the manuscript. Underlying data files can be obtained from the corresponding author.

**Acknowledgments:** The authors wish to thank all the adobe practitioners who came out to train on the test walls while also participating in our study, especially Chris Schrager, US Forest Service, and Ramon Madril, NPS—Tumacácori National Historic Park, for leading the training. Iraida Rodriguez, NPS—Southern Arizona office spent much of her time with in-field scanning, and Matthew Guebard, from the same office, gave invaluable advice and comments on the draft. The authors also thank Fred Pierson of the USDA Agricultural Research Service (ARS), Northwest Watershed Research Center, Boise, Idaho, for use of rainfall simulators applied in this study. We also thank Erin Phelps, co-affiliated with University of Arizona (School of Natural Resources and the Environment) and USDA-ARS Southwest Watershed Research Center, for assistance in completing rainfall simulation experiments. Resources for rainfall simulations were provided by the USDA-ARS. The USDA is an equal opportunity provider and employer. Mention of a proprietary product does not constitute endorsement by USDA and does not imply its approval to the exclusion of the other products that may also be suitable).

**Conflicts of Interest:** The authors declare no conflict of interest.

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
