# Peer review of "Modeling Earthen Treatments for Climate Change Effects"

_heritage, doi:10.3390/heritage6050222_

Round 1

Reviewer 1 Report

The paper tested the effectiveness of three common conservation measures for earthen structure sites in preventing precipitation erosion. The experiment is well-structured, the methodology is well described, and the results is clear, providing a basis for decision-making on conservation measures for earthen heritage. In my opinion, some revisions have to be done before this manuscript could be accepted for publication.

1) A slight restructuring of Introductory section might be better. For example, the third paragraph (line44-53) is about the background and effect of weather conditions on adobe architecture, the second paragraph (line38-42) and part of the first are about existing research on wall testing. Maybe this part can be improved by starting broad in the climate change background and then narrow in on the relevant research, and fully cite existing research on the effects of climate change on earthen materials and on effectiveness testing of conservation measures. etc.

2)Line 38-40  Add a reference

3)Line137  ANOVA appears for the first time, it is recommended to write the full name

4) Line 209-216  I guess this is a description of Figure 5 and needs to point here to Figure 5; similarly, indicate that line 209-216 is a description of Figure 4, if so.

5)Some discussions, such as uncertainties of results (e.g. deviations due to the difference between the simulated and natural precipitation) and why cap works better (whether it is related to the angle of the rainfall. e.g. wind-driven rain), and the generalizability of the results (e.g. applicability to other earth-like materials), would be expected, as those important factors may lead to different realistic results.

6)Line 344  How did the 1,730 cm3 come about, it would be better to explain it.

7)Would it be better not to allow line breaks in the middle of words as there are a lot of word connectors in the whole text.

Author Response

Please find attached answers to your comments.

Reviewer 2 Report

Lines 280-282 state that "the conservation treatments in this study are all designed to be sacrificial treatments on top of the historic adobe" and that "the earth mixture used.... has a higher erodibility than the adobe block fabric". Although outside the scope of this study, I think it would be useful to provide some information on the composition of the historic adobe and about the earth used for the conservation treatments.

Author Response

Please find answers to your comments.
